



# Mechanical models to estimate the paleostress state from igneous intrusions

Tara L. Stephens[1], Richard J. Walker[1], David Healy[2], Alodie Bubeck[1], Richard W. England[1]

[1] School of Geography, Geology and the Environment, University of Leicester, Leicester, LE1 7RH, UK
[2] School of Geosciences, King's College, University of Aberdeen, Aberdeen, AB24 3UE, UK

Correspondence to: Tara Stephens (tls15@le.ac.uk)

**Abstract**

Dikes and sills represent an important component of the deformation history in volcanic systems, but unlike dikes, sills are typically omitted from traditional paleostress analyses in
tectonic studies. The emplacement of sheet intrusions is commonly associated with mode I fracturing in a low deviatoric stress state, where dilation is perpendicular to the fracture plane. Many natural examples of sills and dikes, however, are observed to accommodate minor shear offsets, in addition to a component of dilation. Here we present mechanical models for sills in the San Rafael Subvolcanic Field, Utah, which use field-derived measurements of intrusion
attitude and opening angles to constrain the tectonic stress axes during emplacement, and the relative magma pressure for that stress state. The sills display bimodal dips to the NE and SW and consistent vertical opening directions, despite variable sill dips. Based on sill attitude and opening angles, we find that the sills were emplaced during a phase of NE-SW horizontal shortening. Calculated principal stress axes are consistent (within ~4°) with paleostress results
for penecontemporaneous thrust faults in the area. The models presented here can be applied to any set of dilational structures, including dikes, sills, or hydrous veins, and represent a robust method for characterising the paleostress state in areas where other brittle deformation structures (e.g. faults), are not present.

**1   Introduction**

Sills and dikes are traditionally treated as extension fractures with a dilation vector normal to the fracture wall, i.e. they are extension fractures (*Mode I,* e.g. Anderson, 1951). This assumption has important implications for the use of sheet intrusions in constraining tectonic





stress states, because extension fractures dilate in the direction of the minimum compressive
stress ($\sigma_3$); local deflections of the intrusion attitude are commonly inferred to represent local
rotations of the stress axes. This is most commonly attributed to mechanical layering, and the
presence of pre-existing structures (e.g. Rubin, 1995; Gudmundsson, 2002; 2011a; Magee et
al., 2016). This model implies that intrusions can locally propagate out of the regional $\sigma_1$–$\sigma_2$
plane, via Mode I failure of intact rock, or through Mode I dilation of pre-existing structures,
producing intrusions that display variable dilation vectors along a single intrusion. Notably,
many field examples of sills and dikes exhibit near-parallel dilation vectors, regardless of the
intrusion attitude (e.g. Hoek, 1991; Walker, 1993; Airoldi et al., 2011; Martinez-Poza et al.,
2014; Muirhead et al., 2014; Eide et al., 2016; Walker et al., 2017). Intrusions that demonstrate
shear-offset of markers across their margins indicate that during emplacement the dilation
vector was inclined from plane-normal (Muirhead et al., 2014; Stephens et al., 2017; Walker
et al., 2017); this obliquity of opening can be characterised by the opening angle (**Fig. 1**).

Dilated structures have been studied in detail by Delaney et al. (1986), Baer et al. (1994)
and Jolly and Sanderson (1997), who applied mechanical methods to estimate paleostress states
using sheet intrusion attitudes. These mechanical models have been adopted for statistical
constraints on paleostress axes, and the paleostress state, from vein or dike data (e.g. Sato et
al., 2013; Yamaji, 2016). Although several methods exist to determine paleostress axes, and a
state of paleostress via fault, fracture, or dike data, subhorizontal sheet intrusions are typically
omitted from such analyses. Here, we present mechanical models, based on those of Jolly and
Sanderson (1997), to determine paleostress state using the attitude of dilated fracture sets; we
verify this using the measured opening angle of the intrusions. The method is applicable to any
dilated fracture; here we focus on the stress state associated with sill emplacement in the San
Rafael Subvolcanic Field, Utah, and compare these results to fault data in the same area, to
demonstrate the particular importance of subhorizontal igneous intrusions as records of
paleostress.

## 2    Dilation of pre-existing fractures

Dilation of a fluid-filled, pre-existing, planar structure occurs when the fluid pressure ($P_f$)
exceeds the normal stress on the plane ($\sigma_n$; **Fig. 1**) (Delaney et al., 1986; Jolly and Sanderson,
1997). Normal stress is related to the plane attitude ($\theta$), and the maximum ($S_H$) and minimum
($S_h$) principal stresses acting on the plane (Jolly and Sanderson, 1997):





$$\sigma_n = \frac{S_H + S_h}{2} + \frac{S_H - S_h}{2} cos2\theta, \tag{1}$$

with the range of possible dilatant fracture attitudes controlled by the stress ratio ($\phi$),

$$\phi = \frac{\sigma_2 - \sigma_3}{\sigma_1 - \sigma_3}, \tag{2}$$

and the driving pressure ratio ($R'$) (Baer et al., 1994):

$$R' = \frac{P_f - \sigma_3}{\sigma_1 - \sigma_3}. \tag{3}$$

The stress ratio, $\phi$, is a non-directional value that describes the relative magnitudes of the principal stresses: $\sigma_3 \leq \sigma_2 \leq \sigma_1$ (here, compressive stresses are positive). The driving pressure ratio $R'$ describes the relative magnitudes of the fluid pressure and the remote stress state: When $R' < 0$ (i.e., $P_f \leq \sigma_3$) there is no dilation, and when $R' \geq 1$ (i.e., $P_f \geq \sigma_1$) all fracture attitudes

could dilate. This assumes that an intrusion is emplaced during a deviatoric stress state (i.e. where $\sigma_1 \neq \sigma_2 \neq \sigma_3$); whereas, if $P_f > \sigma_3$ during a non-deviatoric stress state (i.e. where $\sigma_1 = \sigma_2 = \sigma_3$), all pre-existing fracture attitudes could dilate.

Pre-existing fractures, with poles parallel to the $\sigma_3$ axis, will show plane-normal dilation (i.e. dilation parallel to the normal stress vector). Fractures inclined to the plane of $\sigma_3$ resolve

a shear stress on their surface (Delaney et al., 1986). During dilation, shear stress is reduced to zero through a plane-oblique dilation vector (extensional shear). The angle between the dilation vector and normal stress is defined as the opening angle ($\mu$; **Fig. 1a-c**) and represents the ratio of shear to dilation (Delaney et al., 1986). Extensional shear therefore acts to reduce the amount of dilation. For an intrusion comprising inclined and sub-horizontal sections, the inclined

sections will therefore be thinner than sub-horizontal sections oriented perpendicular to $\sigma_3$ (e.g. **Fig. 1c, d**) for a given fluid pressure, providing the stress state is deviatoric and $0 < R' < 1$ (England, 1988). The dilation vector can be measured using traditional compass techniques in the field based on offset piercing points or unique contact geometries, such as recognisable corners in the intrusion walls (e.g. **Figs 1c, 2a, b**). True offsets result from extensional shear

opening **(Fig. 2c, d)**, however apparent shear offset can be produced by marker units that are oriented at an oblique angle to the intrusion plane **(Fig. 2e)**; or via dilation of pre-existing faults, which produce a larger than expected opening angle **(Fig. 2f)**. To determine whether offset and the opening angle is true, or apparent, multiple manual measurements of opening angle and intrusion thickness must be made along strike of an intrusion **(Figs 1c, 2g)**. The opening angle



(μ) of a dilated fracture can be calculated as the inverse cosine of the true thickness (*t*, normal to the plane) and the vertical thickness (*tv*, parallel to the dilation vector; **Fig. 1c**):

$$\mu = \cos^{-1}(t/tv). \tag{4}$$

Alternatively, μ can be measured as a rake of the obtuse angle between the intrusion contact and the dilation vector, minus 90°. Importantly, this measurement only accounts for the
opening angle in two dimensions, movement in or out of the face will not be recorded; therefore dip-parallel sections should ideally be used.

The opening angle is related to the shear stress ($\tau$), normal stress ($\sigma_n$), and fluid pressure ($P_f$) acting on that plane at the time of intrusion (Delaney et al., 1986; Jolly and Sanderson, 1997):

$$\mu = \tan^{-1}\left(\frac{\tau}{P_f - \sigma_n}\right). \tag{5}$$

Equation 5 shows that if the fluid overpressure ($P_f - \sigma_n$) is equal to the shear stress, the opening angle is 45°, and dilation is equal to shear displacement. If the overpressure is greater than the shear stress, the opening angle is less than 45°, and the fracture will show a greater component of dilation to shear. When μ is negative, the fracture will remain closed as the fluid pressure
did not exceed the normal stress. An intrusive segment, however, may inflate against a closed fracture (where $P_f < \sigma_n$), causing a local contractional shear, and a blunt intrusion tip (e.g. **Fig. 2g**).

## 3    Mechanical models for fracture dilation and opening angle

Equation 5 can be visualised in three dimensional (3D) space, for any given stress state, stress ratio and fluid pressure. Figure 3 shows the opening angle (μ) of all possible fracture planes in 3D space, plotted as colour-contoured pole to plane values of μ, on equal area lower hemisphere stereographic projections and on 3D Mohr Circles. Fluid pressure was calculated at five equal intervals relative to the ambient stress state, which are expressed as *R'* values (**Fig. 3a**).
Opening angles are plotted for three tectonic regimes where $\phi$ is 0.5: (1) thrust (**Fig. 3b**); (2) strike slip (**Fig. 3c**); and (3) normal faults (**Fig. 3d**). The principal stress attitudes are constant in all models, with the azimuth for maximum horizontal compression ($\sigma_H$) trending E-W. All planes are modelled as cohesionless surfaces.





The models complement the results of Jolly and Sanderson (1997), and demonstrate that
within a given stress state, increasing the fluid pressure increases the range of fracture attitudes
that can dilate. The models also show that for any given fracture, increasing $P_f$ decreases $\mu$.
Where $\sigma_3 < P_f \leq \sigma_2$ (**Fig. 3b-d, i–ii**), the dilation zone, delineating the poles to fractures that are
predicted to dilate, forms an ellipse about the $\sigma_3$ axis and is elongate in the direction of the
intermediate stress ($\sigma_2$); only fractures with poles parallel to the $\sigma_3$ axis show Mode I opening.
If $\sigma_2 < P_f < \sigma_1$ (**Fig. 3b$_{iii}$-d$_{iii}$**) the dilation zone forms a girdle parallel to the axis of $\sigma_2$, with two
defined zones of near Mode I opening (0-10º) surrounding the $\sigma_2$ and $\sigma_3$ poles. Fractures of all
attitudes will dilate if $P_f = \sigma_1$ (**Fig. 3b$_{iv}$-d$_{iv}$**). The models suggest that Mode I opening of pre-
existing fractures should only be common if the fluid pressure exceeds $\sigma_1$ (e.g., **Fig. 3b$_v$-d$_v$**). It
is important to note that pre-existing fractures must become linked to the magmatic system in
order for them to be dilated and intruded; where the normal stress acting on a fracture, which
is in contact with an intrusion, exceeds the fluid pressure, the fracture will not dilate, but the
intrusion may inflate against it, creating a blunt tip ($\mu=90°$) (Stephens et al., 2017). Therefore,
by measuring the opening angles of veins or intrusions, and determining the distribution of
fracture attitudes that are in contact with intrusions but were not dilated, it is possible to
constrain the fluid pressure relative to the ambient stress state. Here, we have applied this
method to igneous sills in the San Rafael Subvolcanic Field, Utah (SRSVF).

Dilated fractures, created during the same dilational event, will produce a unique
distribution of opening angles when plotted stereographically as poles to planes (e.g. Jolly and
Sanderson, 1997; Yamaji et al., 2010; Stato et al., 2013). Whether this is a tight cluster or a
distributed set of poles, the pattern is usually interpreted to represent a single event, governed
by one fluid pressure (e.g. Jolly and Sanderson, 1997); however, it is possible that the data
represents multiple events caused by fluid pressure pulses of varying magnitude (Yamaji et al.,
2010). As the opening angle represents the ratio of shear to dilation (Delaney et al., 1986),
magmatic events of varying magnitude, within the same governing stress state, could produce
cross-cutting intrusions with similar geometries, but different opening angles (e.g. **Fig. 3**). The
attitude and opening angle of dilated fractures, along with cross-cutting relationships, can
therefore be used to identify whether a network of dilated fractures represents one, or multiple,
fluid pressure pulses and the stress state during each pulse.



## 4    Stress state model for sill emplacement: San Rafael Subvolcanic Field

The San Rafael Sub Volcanic Field (SRSVF) is located in the north-western Colorado Plateau, central Utah (**Fig. 4a**), and comprises numerous, sills, dikes and volcanic breccia bodies (**Fig. 4b, c**); no crustal magma chambers have been identified as the source for the intrusive complex (Delaney & Gartner, 1997). The exposed intrusive system was emplaced at ~1 km depth, within the Mid Jurassic sedimentary rocks of the San Rafael Group, which comprises four formations; the Carmel Formation (siltstone and shale), Entrada Sandstone (interbedded sandstone, siltstone and claystone), Curtis Formation (interbedded glauconitic sandstone, and siltstone), and the Summerville Formation (siltstone, sandstone and shale) (Gartner, 1986). The sequence represents a paralic environment, with near-shore and shallow marine deposits (Gartner, 1986; Peterson, 1988). The sills are mainly emplaced within the Entrada Sandstone, however they also cut across formation boundaries to intrude the Carmel and Summerville Formations (e.g. **Fig. 4d**; Walker et al., 2017). Sills and dikes display mutual cross-cutting relationships, but no feeding relationships have been observed (e.g. **Fig. 4e**; Gartner, 1986; Walker et al., 2017). Delaney and Gartner (1997) determined ages of dike emplacement between 4.6 and 3.7 Ma; the field relationships therefore constrain sill emplacement to a 1 Myr interval.

Walker et al. (2017) interpret the sills in the SRSVF as low-angle conjugate intrusions, they show that the sills record a phase of horizontal tectonic shortening, rather than relating to local deflections due to material layering. The sills range from <10 cm to ~30 m thick, and are discordant to bedding, with dips of 1 – 25° (**Fig. 4d, f**). Based on sill attitude measurements over km-scale outcrops, Walker et al. (2017) showed that sill poles cluster about a near-vertical axis with two defined clusters: a NW and SE dipping set in the northern SRSVF, and a NE and SW dipping set in the southern SRSVF. This study focuses on the NE and SW dipping sills in the southern SRSVF (**Fig. 4b, c**).

Across all scales, we find mutual cross-cutting relationships between sills with bimodal dips (e.g. **Figs 1c, 5a-c**), and en echelon sill segments which define low-angle (~20° dip) conjugate sets (**Fig. 5a**). Deformation bands and gypsum veins have comparable attitudes to sills and thrust faults in the field area, and define low-angle conjugate sets with NE and SW dips (**Fig. 5d-i**). Sills intrude, and are cut by, low angle fractures, thrust faults and reverse faults (all referred to as thrust faults from hereon in), and either cut or abut against sub-vertical fractures (**Figs 4d, 5f**; Walker et al., 2017). A constant near-vertical opening direction is noted across all sill segment attitudes (**Figs 6, 7a**). Local sill contact data displays a range of dips from (4° - 87°) with horizontal to subhorizontal sill segments (0-20° dip) showing plane-normal





(Mode I) and extensional shear dilation (μ = 0-20°), respectively (**Fig. 6a, 7a**). Inclined segments (20-56° dip) are consistently thinner than adjoining horizontal segments and display

reverse-sense shear offset of piercing points ($20° < μ \lesssim 75°$; **Fig. 7a**). Subvertical to vertical fractures (>70º dip) were not intruded (**Fig. 7a**) suggesting that fluid pressure was less than the normal stress on those planes during intrusion, and the opening angle was < 0°.

Since the range of dilated sill attitudes is known, we can determine the relative stress and fluid pressure, using the parameters $\phi$ and $R'$. We calculate these parameters

stereographically, using an amended method from Jolly and Sanderson (1997). The local sill contact data are plot as poles to planes and coloured by their μ-value; the pole cluster showing Mode I opening (μ=0-10º) is fit with an ellipse (**Fig. 7b**). For data with a clustered distribution, this ellipse geometry provides a guide for the total range of dilated fractures, where blunt tips define the limit of fracture attitudes able to dilate; beyond this is the zone of no opening (where

μ<0º; **Fig. 7b**). The minimum compressive stress ($σ_3$) plots in the centre of the Mode I ellipse (which likely coincides with the centre of the data cluster); $σ_1$ and $σ_2$ are mutually orthogonal to this (**Fig. 7c**). If the data has a girdled distribution, the 0-10° ellipse should be used as an approximation for the geometry of the zone of no opening. In these cases, there will be two clusters of poles where μ=0-10° (see **Fig. 3biii-diii**), the larger ellipse signifies the location of

$σ_3$. A distributed set of dilated fractures may contain several zones of Mode I opening (μ=0-10°; see **Fig. 3**): $σ_3$ will plot in the larger of these ellipses and $σ_1$ in the zone of highest opening angle, or the smallest Mode I ellipse when $P_f$ exceeds $σ_1$ (see **Fig. 3bv-bv**).

Field data for sills in the SRSVF can be fitted to an elliptical region on the stereonet with a NW-SE long axis (**Fig. 7b**), giving horizontal NE-SW maximum compression where:

$σ_1$ plunges 3° towards 068°; $σ_3$ is vertical, plunging 87° towards 265°; and $σ_2$ is horizontal plunging 1° towards 158° (**Fig. 7c**). Using the stereonet data in **Figure 7c** we derive the angles $θ_1$ and $θ_2$ which are used to calculate the stress ratio ($\phi$) and driving pressure ratio ($R'$) by Jolly and Sanderson (1997):

$$\phi = \frac{1+\cos(2θ_2)}{1+\cos(2θ_1)}, \tag{6}$$

and

$$R' = \frac{1+\cos(2θ_2)}{2}, \tag{7}$$



where $\theta_1$ is the angle between the $\sigma_2$ axis and the perimeter of the dilational ellipse and $\theta_2$ is the angle between the $\sigma_1$ axis and the dilational ellipse (Jolly and Sanderson, 1997). When an ellipse can only be fit to the zone of no dilation, Jolly and Sanderson (1997) provide an

215 alternative method for calculating $\phi$. The $R'$-value is two dimensional (2D); it does not take into account the magnitude of $\sigma_2$. The three dimensional relationship between the fluid pressure and all principal stresses is illustrated by the ellipse geometry on the stereonet (**Fig. 7c, d**), and can also be visualised through construction of a 3D Mohr circle (**Fig. 7e**).

The calculated $\phi$ (0.77) and $R'$ (0.68) values define the stress ratio and driving stress
used to create the opening angle mechanical model (**Fig. 7d**). For the model, we assigned a minimum, vertical stress of 25 MPa to simulate a ~1 km emplacement depth (Gartner, 1986). Development of extensional shear fractures requires a low differential stress ($\sigma_D$: $\sigma_1 - \sigma_3$): $4T > \sigma_D < 5.66T$; compressional shear faults require that $\sigma_D > 5.66T$ ($T$ is the host rock tensile strength; Sibson, 2003). Due to the bimodal (conjugate) dip distribution of the sills, their
consistent near-vertical opening direction, and the mutual cross-cutting and intrusive relationship between thrust faults and sills we estimated that $\sigma_D = 6T$. We estimate the tensile strength of the host rock at 1 km depth to be 3 MPa, giving $\sigma_D$ = 18 MPa; though due to the nature of the model, providing all parameters are scaled relative to $T$, the value of $T$ does not change the resulting opening angle pattern, only the relative magnitudes of the principal
stresses and fluid pressure. $\phi$ (0.77) and $R'$ (0.68) derived in the model indicate a mild horizontal radial compression during emplacement, with fluid pressure less than $\sigma_2$ (**Fig. 7d, e**).

The measured opening angles of the studied sills fit with the contouring of a single ellipse (**Fig. 7d**), suggesting that during the intrusion of the sills in the study area, the fluid
pressure and stress state remained relatively constant; however, we acknowledge that a broader study across the SRSVF may reveal fluctuations in either stress state or magma pressure.

## 5    Sill geometry as a record of stress

Intrusions are typically treated as hydrofractures that formed in a low deviatoric far-field stress
state (i.e. where $\sigma_1 \approx \sigma_2 \approx \sigma_3$), where the magma pressure exceeded the minimum compressive stress plus the host rock tensile strength (i.e. $P_f \geq \sigma_3 + T$; e.g. Gudmundsson, 2002; Kavanagh



et al., 2006). For such a case, it should be possible to dilate pre-existing cohesionless fractures of all attitudes in a Mode I sense, and fractures of all attitudes should have similar apertures (true thickness). In a deviatoric stress state (i.e. where $\sigma_1 \neq \sigma_2 \neq \sigma_3$), the aperture of a fluid-

filled fracture is influenced by its attitude relative to the principal stress axes, the fluid overpressure ($P_f - \sigma_n$) within the fracture, and the host rock elastic properties (Young's modulus, $E$, and Poisson's ratio, $\nu$), at the time of dilation (e.g. England, 1988; Gudmundsson, 2011b). As noted in Section 2, fractures that are oriented parallel to the $\sigma_1$-$\sigma_2$ plane (perpendicular to the $\sigma_3$ axis) will have a greater thickness than those oriented obliquely to the

$\sigma_1$-$\sigma_2$ plane (for a given fluid pressure), as the fluid overpressure required to cause dilation must also relax the shear stress on an inclined surface. Material elastic properties control how the host rock responds to an applied stress: Units with high $E$ are less compressible than those with low $E$. Therefore, if a fluid-filled fracture with constant fluid pressure, cross-cuts units with varying elastic properties, the fracture will have a larger aperture in units with low $E$, and a

smaller aperture in units with higher $E$ (Gudmundsson, 2011b). The studied sills in the SRSVF predominantly intrude Entrada sandstone units and display along-strike changes in attitude; thinner inclined sections and thicker horizontal sections (**Figs 5a-c, 6, 7a**). These variations of attitude and thickness do not correspond to bedding interfaces, or lithological changes in the host rock. Sharp contacts between cross-cutting sills suggest that older sills were solidified

prior to intrusion of the younger sills (**Fig. 6**). As such, contrasts in elastic properties between the older sills and the sandstone could have promoted thickness variations in the younger sills, however, where sills are cross-cutting, the younger sills maintain a constant thickness when they are near-horizontal; thickness changes are only associated with inclined sections (**Fig. 6**). We infer that for the sills in the SRSVF detailed in our study, along-strike thickness variations

are controlled by the segment attitude, the far-field stress state, and the fluid pressure, rather than host rock elastic properties.

Variations in the intrusion attitude are commonly thought to represent local rotations of the principal stress axes. This is based on the assumption that dilated fractures form via Mode I failure in low deviatoric stress states where $\sigma_D < 4T$, and the intrusion lies in the $\sigma_1$-$\sigma_2$

plane, and normal to the $\sigma_3$ axis. Hence, areas that host several intrusion sets are interpreted to represent discrete and separate intrusive events (e.g., Delaney and Gartner, 1997). This applies particularly to intrusions that form as a result of magma chamber overpressure, where $\sigma_1$ stress trajectories emanate away from a magma chamber with an approximate conical geometry





(Gudmundsson, 2006; Martí and Geyer, 2009). Whether the $\sigma_1$ trajectories are low-angle (sill-like) or high angle (dike-like), they can be locally modified by pre-existing structures and host rock mechanical properties (Gudmundsson, 2006). The radial distribution of the $\sigma_1$ trajectories implies that sheets of opposing dip direction that form during the same magmatic event cannot cross-cut each other. Sills in the SRSVF, however, display bimodal (conjugate) dip patterns, across the metre- to kilometre-scale, with mutual cross-cutting relationships and consistent vertical opening directions, indicating that these sills *are not* cone sheets, and they *were not* the result of local stress reorientations. Our study suggests that intrusions of various attitudes may form *simultaneously* (i.e. within the same magmatic episode), as a result of emplacement coeval with a high deviatoric far-field stress state (e.g., Walker, 2016; Walker et al., 2017). This has particularly important implications for regions that may not exhibit mesoscopic or larger deformation features (e.g., faults and fractures), other than intrusions, from which the paleostress may be derived.

The SRSVF is host to dikes and sills, and these sills cut, and are cut by, thrust (and reverse) faults that dip to the NE and SW (**Figs 5f, 8a**; Walker et al., 2017). The thrusts, deformation bands and gypsum veins form conjugate sets (**Fig. 5d-i**), which record a coaxial horizontal shortening and vertical thickening (Walker et al., 2017). Right Dihedra paleostress analysis (Delvaux and Sperner, 2003) of the thrust and deformation band data for the San Rafael shows a close correlation with the derived stress state results for sills, with an angular mismatch between the principal stress axes of ~4º, and $\phi$-values of 0.53-0.56 (**Fig. 5h, i**). For comparison, Bingham analyses (Yamaji, 2016) of the local sill contact data derives an average $\phi$-value of 0.63 and principal stress axes creating a 6º angular mismatch with the opening angle and Right Dihedra models (**Fig. 8b**). To further constrain the stress ratio, and the relationship between intrusions and contractional shear structures, we input the thrust fault, deformation band, gypsum vein, and overall sill geometry pole data into mechanical models of normalized slip tendency ($T_s$; **Fig. 8c**) and dilation tendency ($T_d$; **Fig. 8d**). Although typically used to assess the reactivation potential of pre-existing structures in a present-day stress state (e.g. Ferrill et al., 1999), the models can also be used to fit a paleostress state to field data (e.g. Stephens et al., 2017). Normalized slip tendency ($T_s = (\tau / \sigma_n) / T_s max$; Morris et al., 1996) and dilation tendency ($T_d = (\sigma_1 - \sigma_n) / (\sigma_1 - \sigma_3)$; Ferrill et al., 1999) are calculated from the stresses acting on a plane, or potential plane. This can be used to predict the attitude of both potential failure planes, and pre-existing fractures that are susceptible to reactivation in a given stress state. Notably, the zone of high $T_s$ (0.8-1.0) overlaps with the zone of high $T_d$ (0.8-1.0), indicating a



likely zone of extensional shear. To enable conjugate shear failure, we used the same differential stress of $6T$, where $T = 3$ MPa, as used previously. By fitting the zones of high slip tendency to the combined thrust fault and deformation band data, and zones of high dilation tendency to the overall sill geometry and gypsum vein data, we were able to derive a best-fit $\phi$-value of 0.65, with a horizontal NE-SW $\sigma_1$ that plots within the 6° angular mismatch. The $\phi$-value is consistent with the Right Dihedra ($\phi$=0.53; **Fig. 5h, i**), Bingham analysis ($\phi$=0.63; **Fig. 8a**) and our opening angle models ($\phi$=0.77; **Fig. 7**). The combined field observations and paleostress analyses suggests that sill emplacement took place during a state of mild horizontal radial compression.

A key difference between the strains recorded by sills and thrusts is the dominance of dilation during sill emplacement, and compressional shear during thrusting. The range of local dips, and consistent vertical opening, suggests that the sills were emplaced by a combination of brittle failure and dilation of pre-existing structures (including thrust faults); where the fluid overpressure accommodates both shear and dilational strains. Consistent with the interpretation of Walker et al. (2017), our field observations and mechanical model results suggest that the sills represent conjugate intrusions, which record the continuity of horizontal shortening during periods of elevated magmatic pressure.

While the opening angle of dilated fractures has been used here to characterise the paleostress state during dilation of *pre-existing* fractures, we envisage that it could also be applied to tensile or extensional shear fractures formed via failure of intact rock. In these cases, we would expect the fracture network to comprise of predominantly parallel, or low-angle bimodal (conjugate) intrusions, with opening angles of <20°. Failure of intact rock requires a higher fluid pressure than for reactivation of pre-existing structures. As such, although our models calculate the stress state and stress ratio, the derived estimates of driving fluid pressure ratio would be minimum values, and not truly representative of the fluid pressure required for emplacement via intact rock failure. Importantly, the models can be used to determine whether intrusive suites record changes in the tectonic stress state, or to identify fluid pressure pulses of varying magnitude in a single governing far-field stress state. These attributes have significant implications for improving our understanding of the development of past and present-day magmatic systems.



## 6    Conclusions

Our mechanical models build upon the work of Delaney et al. (1986) and Jolly and Sanderson
(1997) to use fracture geometry and opening angles to derive the principal stress axes during
sill emplacement, and provide crucial new constraints on the stress state and fluid pressure,
applicable to dikes, sills, and veins. The geometry of sills in the SRSVF record a continuous
deformation that is otherwise accommodated by contractional faults in the area. Contoured
regions on stereonets for opening angles suggest that in a high deviatoric stress state, it should
be relatively rare for intrusions to be purely Mode I structures. In such settings, it should be
commonplace for intrusions to accommodate a component of shear during their emplacement.
Our opening angle model is particularly useful in determining paleostress states for regions
where there is little brittle deformation (i.e. faulting), other than intrusions, and it may therefore
present a useful and important tool in tectonic and magmatic studies.

**Acknowledgements**

This work was undertaken during T.L. Stephens's PhD studentship, supported by the Central
England Natural Environmental Research Council (NERC) Training Alliance (CENTA)
[award reference: 1503848]. The authors would like to thank Craig Magee, Atsushi Yamaji and
an anonymous reviewer for their constructive reviews on an earlier version of this paper.





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

465



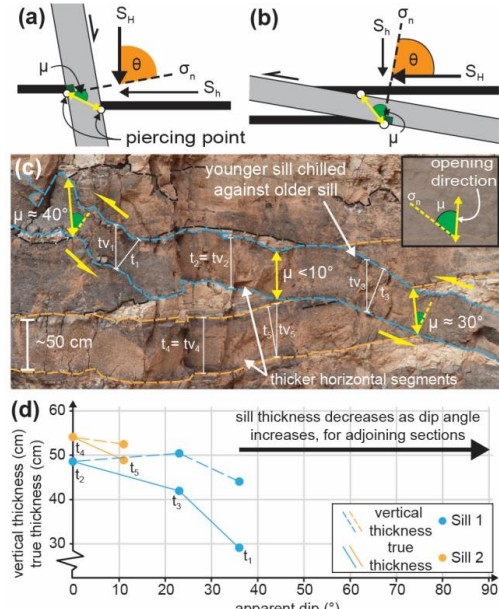

**Figure 1.** Schematic diagrams to show the dilation direction (yellow arrow) and opening angle (μ) of fractures in different stress states: **(a)** normal fault regime (where $S_H = \sigma_v$); **(b)** thrust fault regime (where $S_h = \sigma_v$); the plane attitude is described by θ, the angle between the normal stress and the maximum stress. **(c)** Example of cross-cutting sills; the middle sill (outlined in blue) shows a consistent vertical opening direction: Mode I opening on horizontal planes, and extensional shear opening on inclined planes. **(d)** Graph showing changes in vertical thickness, tv, and true thickness, t, with changes of apparent dip, for the sills shown in (c).



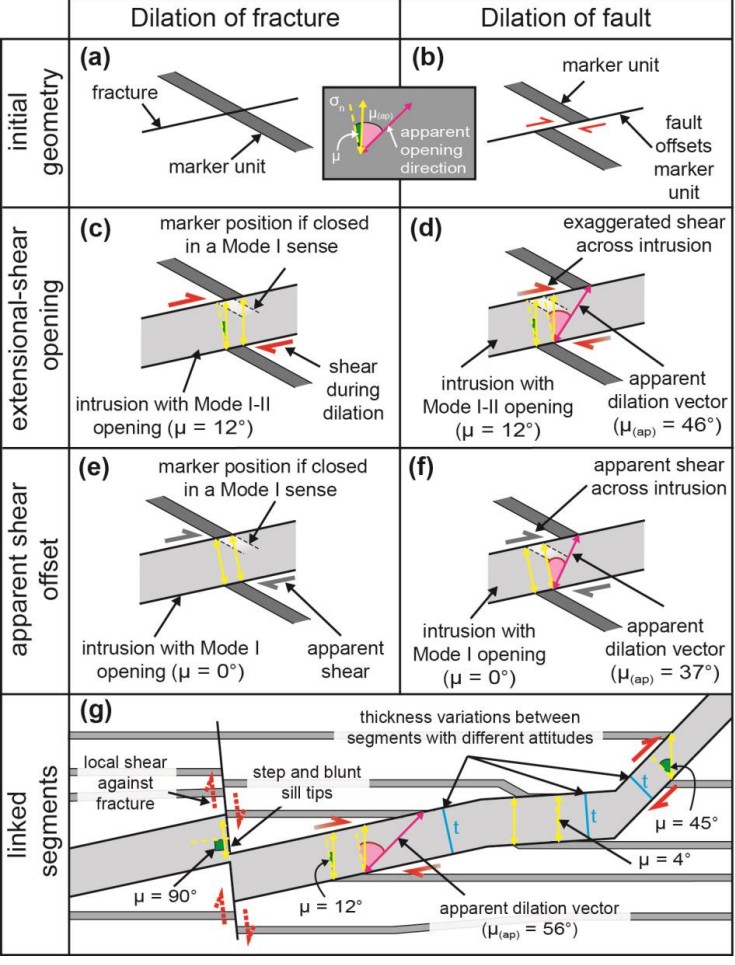

**Figure 2.** Figure 2. Schematic diagrams to show the relationship between opening mode and apparent shear. **(a-b)** Initial pre-intrusion geometries with a marker unit oblique to the fracture/ fault plane. **(c-d)** Extensional shear dilation of a pre-existing fracture and fault. **(e-f)** Mode I dilation of a pre-existing fracture and fault, creating an apparent shear offset. **(g)** Dilation directions and opening angles along linked segments can be used to infer the true dilation direction and fluid pressure during intrusion. μ(ap) = apparent opening angle.





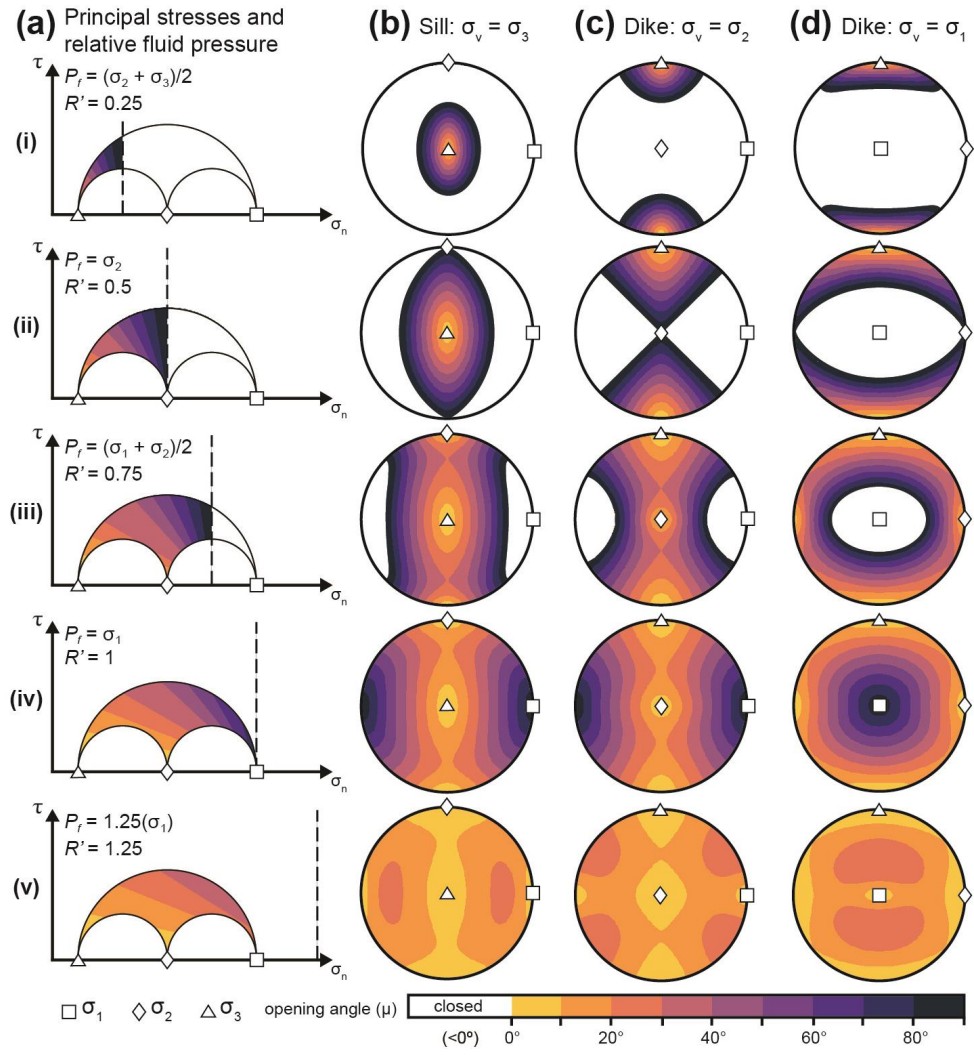

**Figure 3.** Mechanical models showing contoured fracture-opening angles for various stress and fluid pressure conditions, using the equation from Delaney et al. (1986), projected onto lower hemisphere, equal area stereonets and Mohr Circles (colour scheme from Thyng et al., 2016). Five increments of fluid pressure (**a$_{i-v}$**) have been modelled with three stress states: **(b)** Thrust fault regime; **(c)** strike-slip regime; **(d)** normal fault regime. The dashed line on the Mohr Circles indicates the fluid pressure magnitude.



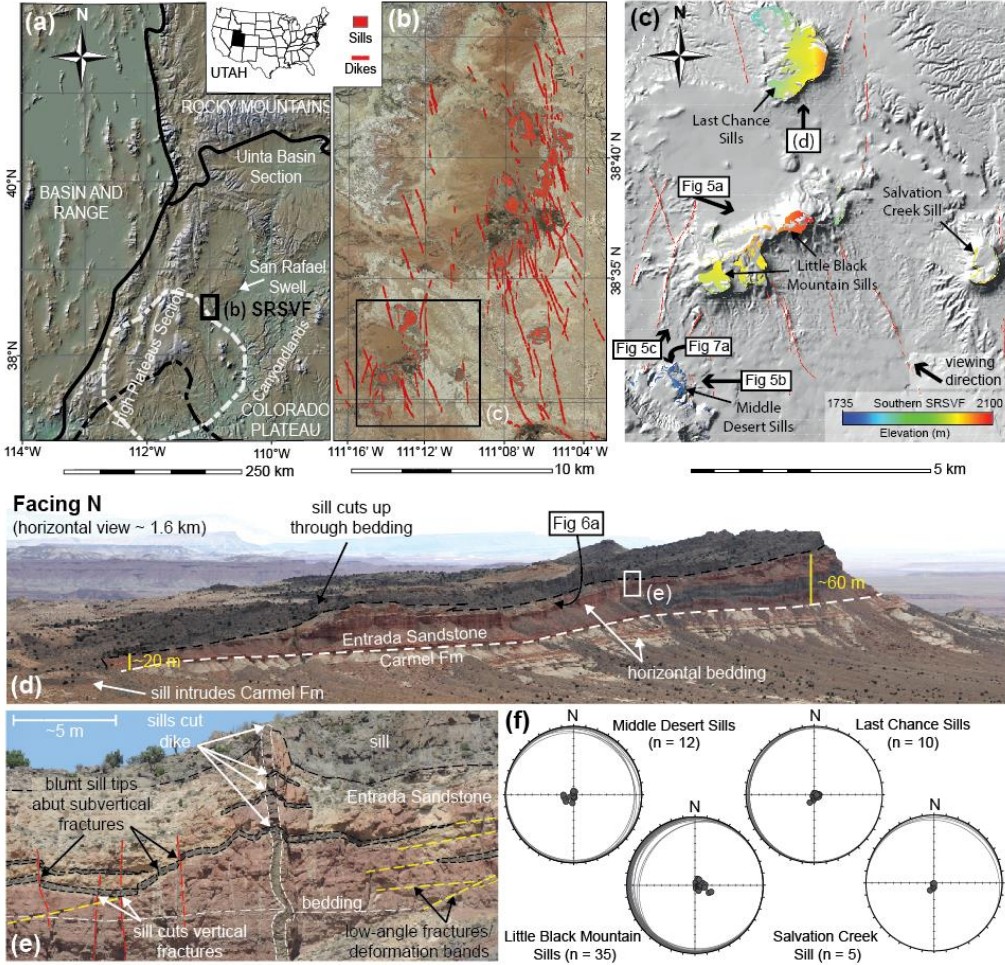

**Figure 4.** Location maps for the San Rafael Sub-Volcanic Field in Utah amended from Walker et al. (2017). **(a)** Digital elevation Model for Utah, showing major structural and depositional areas of the Colorado Plateau. Solid black line shows province boundaries. Dashed black line is a region of lower-crustal delamination and crustal thinning detailed in Levander et al. (2011); dashed white line is their outline of a downwelling body at 200 km depth, estimated from body wave tomography. **(b)** Aerial imagery for the San Rafael Sub-Volcanic Field (SRSVF) highlighting location and distribution of intrusive bodies. **(c)** Hillshaded digital elevation models for thick sills in the southern SRSVF, coloured to show extrapolated elevation data for sill top contacts (modified from Walker et al., 2017). **(d)** Field photograph of the Last Chance Sills, which transgress through the stratigraphy; the upper sill cuts across the Carmel-Entrada Formation boundary. **(e)** Sills cut dike and cut, and abut against, subvertical fractures (sills outlined in black, dike in white). **(f)** Lower hemisphere stereographic projections show sill top contact polygon attitudes (extrapolated from c) as great circles, and poles to planes for each sill system (data from Walker et al., 2017).





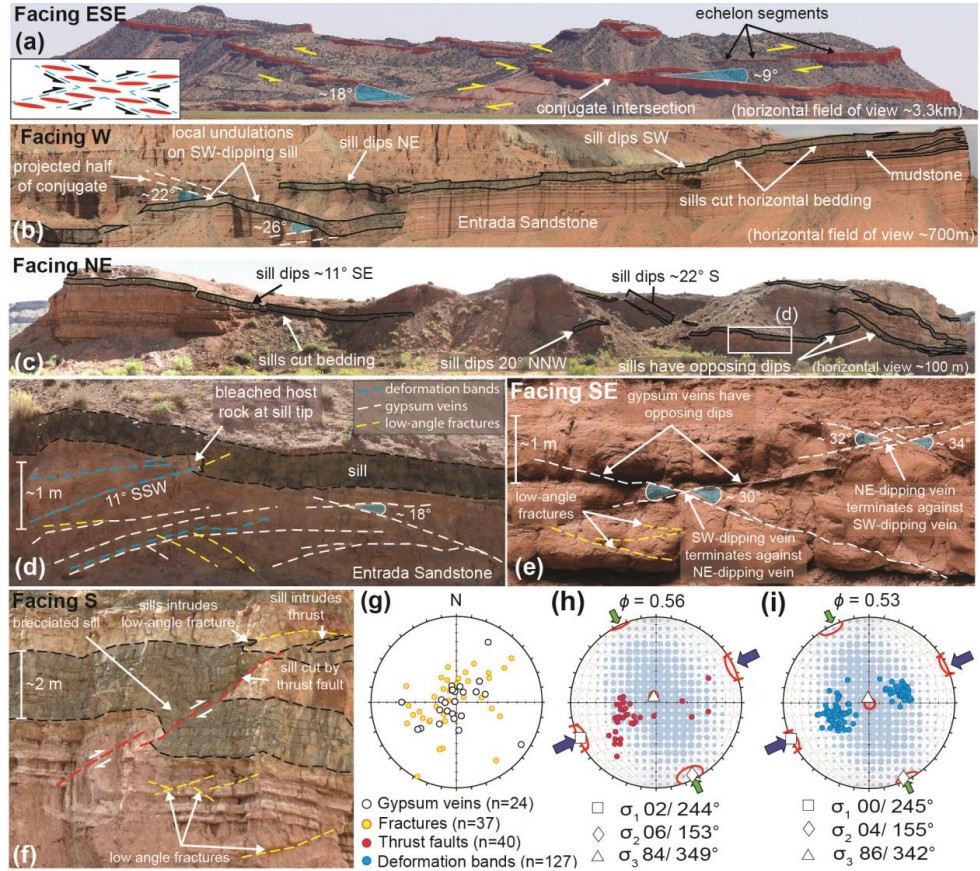

**Figure 5.** Sill geometry and paleostress analyses of deformation structures in the San Rafael. **(a)** km-scale segmented sills show en-echelon stepping consistent with conjugate faults (sills shaded red). Inset shows schematic interpretation. **(b)** 100 m scale sills showing NE and SW dips (sills outlined in black). **(c)** <100 m scale sills with bimodal dips (sills outlined in black). **(d)** Deformation bands and low-angle fractures parallel gypsum veins. **(e)** Gypsum veins with conjugate geometries. **(f)** Sill intruding and cut by a thrust fault and low angle fractures (sills outlined in black). **(g-i)** Lower hemisphere equal-area projections, showing: **(g)** gypsum veins and fractures, **(h)** thrust faults, and **(i)** deformation band data; principal stress axes calculated for **(h)** and **(i)** using the Right Dihedron method (Delvaux and Sperner, 2003). Calculated stress axes given as plunge/ trend measurements, $\phi$-value is the stress ratio ($\sigma_2$-$\sigma_3$/ $\sigma_1$-$\sigma_3$).





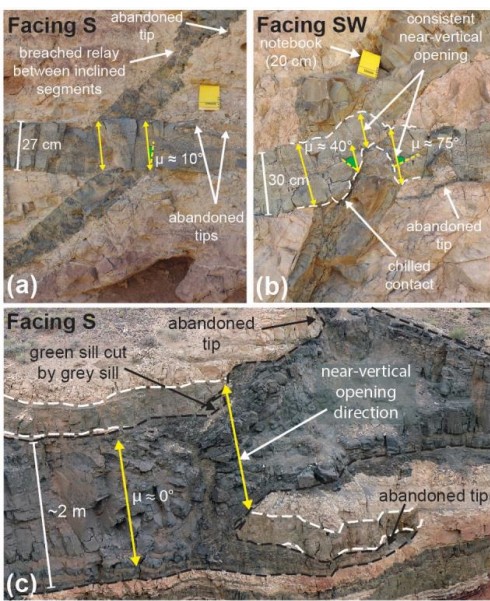

**Figure 6.** Sill-sill cross-cutting relationships. **(a)** Younger horizontal sill maintains a constant thickness when cutting older sill. **(b-c)** Inclined sill sections are thinner than horizontal sections. All sills show near-vertical opening directions, regardless of attitude.



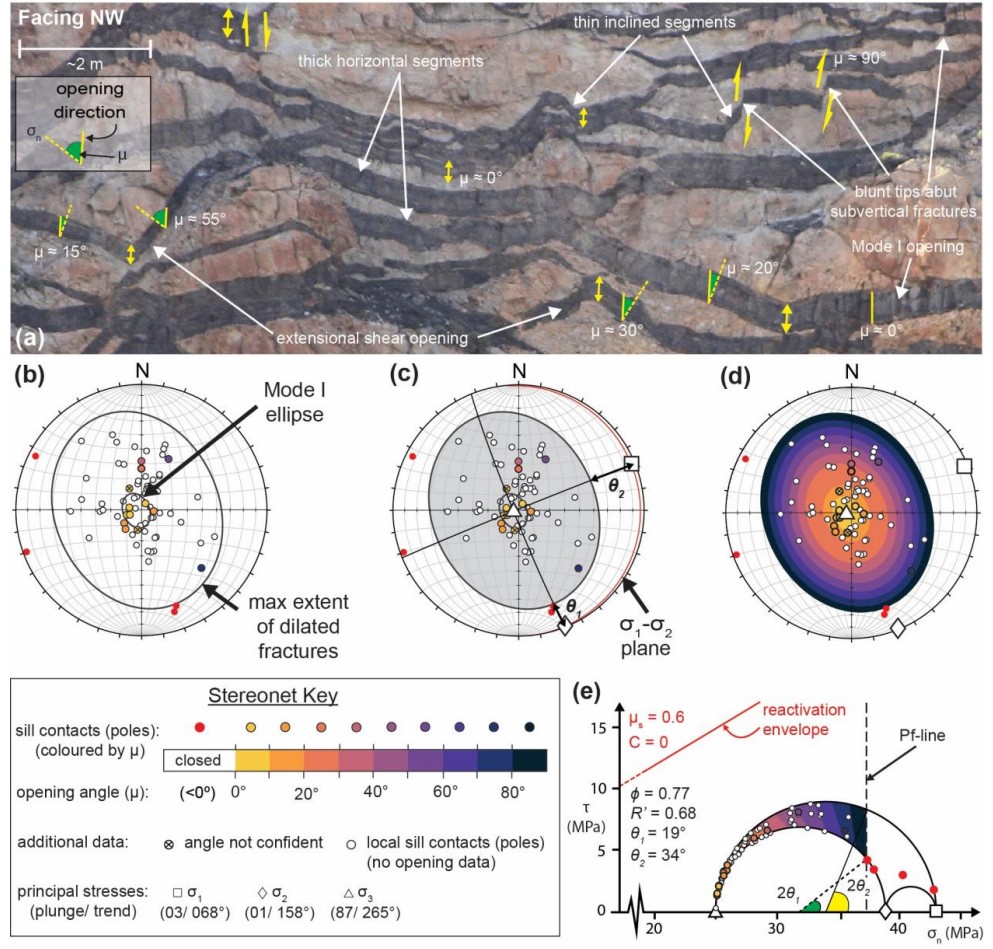

**Figure 7.** The opening-angle mechanical model. **(a)** Field example of thin sills (<1m thick) displaying a range of local contact dips, and a consistent vertical opening. **(b)** Lower hemisphere, equal area stereonet with local sill contacts plotted as poles to planes, coloured relative to their opening angle, with Mode I and total dilational ellipses constructed. **(c)** Determination of principal stress axes and $\theta$ angles. **(d)** Mechanical model results using the stress ratio and driving pressure ($\phi$=0.77, $R'$=0.68; see text for details) derived from **(c)**. **(e)** 3D Mohr circle showing how $\theta_1$ and $\theta_2$ can be used to calculate relative fluid pressure; Mohr Circle is colour contoured for values of $\mu$, local sill contacts are plotted as poles and coloured by their opening angle.





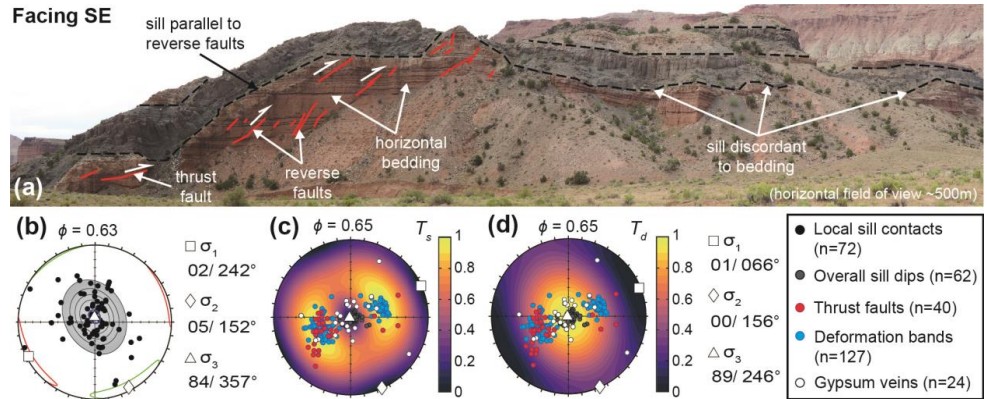

**Figure 8.** Sill geometry and paleostress analyses of deformation structures in the San Rafael. **(a)** Field example for comparison between sill and thrust fault geometry. **(b-d)** Lower hemisphere equal-area projections with calculated stress axes given as plunge/ trend measurements: **(b)** Local sill-host rock contact data, Bingham analysis (Yamaji et al., 2016) used to calculate principal stress axes, shown with 95% confidence regions and density contours (shaded region). **(c)** and **(d)** are mechanical models contoured for slip tendency ($T_s$) and dilation tendency ($T_d$), respectively.