# Peer review of "Mechanical models to estimate the paleostress state from"

_Solid Earth, 2018_

## Referee Comment (RC1) · Anonymous Referee #1 · 17 Apr 2018

This paper is interesting and well written, providing a mechanical model for using sill intrusion attitudes and opening vectors to constrain magma pressures and stress states during emplacement events. Admittedly, the approach and model is not completely new, as it builds off original work from authors such as Jolly and Sanderson (1997). However the study is timely and provides an advance in how we view sill intrusion systems generally.

There is currently a predominant view in the scientific literature that sill intrusions result primarily as a result of bedding and layer heterogeneities. Although this appears to be the case for some examples, it is equally possible that many sill systems form in compressional settings, where regional least compressive stress is vertical. Furthermore, dike intrusions are frequently used as paleostress indicators. The same should ap-

ply to sills also, particularly when accompanied by careful and detailed field kinematic inspections, as has been applied in this study.

Recent work from this group has already applied sills as paleostress indicators, and this paper focuses largely on detailing the methodology for such an approach, whilst providing a nice field dataset for proof of concept. It is a timely piece of science, which will benefit future structural, tectonic and volcanological studies investigating tectonic-magmatic interactions.

Below I provide some general and specific comments. The paper is in an excellent state and appears to have gone through a number of reviews already, so to me the manuscript requires only minor corrections. I look forward to seeing a final published version of this paper, which I think will provide a useful resource for future intrusion studies.

GENERAL COMMENTS

1) As discussed below, there may be potential for this method to be misapplied in the absence of detailed investigations of the timing and history of intrusive events, and the authors should explicitly discuss the critical field observations needed before applying the approach. The authors have previously done extensive and careful field work in this study area (e.g., Walker et al., 2017), so I believe the approach is valid for the San Rafael field. However, the method assumes that the stress field remains constant during emplacement of the measured intrusions, and thus the magnitude of fluid pressure (i.e., magma) is the critical parameter that determines the spread of the intrusion attitude data. With this in mind, the method also requires the emplacement of the measured intrusions, which exhibit a variety of attitudes, to be closely spaced in time (i.e., L137: "created during the same dilational event"). If one were to take this approach to a different field setting, it therefore may be misapplied if the timing of events is unknown.

So with this in mind; in contrast to the model presented in this study, what if the magnitude of the fluid pressure remained relatively low (thus restricting the range of intrusion attitudes at any given time and place), but the orientation of the stress field varied both temporally and spatially? Such a scenario would seem really likely under a classic cone sheet model for example.

The authors somewhat address this point on L 267-285, but I think a little more discussion is needed so that readers can explicitly see the potential pitfalls with the approach, and be shown how to deal with these with detail field observations and measurements. For example, what is unique about this study, compared to Jolly and Sanderson (1997), is that the authors also present the attitudes of fractures exhibiting compressional shear failure, thereby tightly bounding the ellipse in the stereonets that provide constraints on fluid pressures. The interconnectivity between intrusions of different attitudes is also illustrated, suggesting they formed near-contemporaneously. I feel that careful measurements like these robustly support the model, and future studies could also consider this approach.

2) It would be useful for the reader to also have some quick background on why sills have been omitted from these types of paleostress analyses in the introduction. This could be included as a few sentences immediately after the comments on lines 46-48. To me, this issue stems from the problem that (1) sills in many cases must be fed by dikes, and are often observed in regions which are thought to be extensional. Thus if we consider stress as the primary control on sill orientation, then the least compressive stress must be both vertical and horizontal in the region where sills are fed, which is often in an extensional setting. The effects of mechanical layering essentially act as a work-around for this paradox. (2) Sills are often observed intruding sedimentary layers, although this is no surprise as sedimentary layering is often horizontal. I know this group has brought up points such as these in recent papers, but I feel that it is important to bring these points to the forefront to provide broader context for the paper.

Specific comments

L 115: Please consider the terms "extensional" and "compressional" as the subject matter is intrusions rather than faults.

L 141-143: The other possibility is that it is multiple events where the orientation of the principle stresses has varied. I don't think this is the case in the paper, but it should not be overlooked. See Comment #1.

L 162-163: No direct feeder relationships are observed, which is very rare in nature anyhow, but they have been inferred. For example, Figure 3 of Richardson et al. (2015) points to potential dike feeder based on sill thicknesses. Additionally, the thickness distributions of these sills do suggest potential NNE-SSE-trending feeders, which is parallel to the dikes (i.e., Walker et al., 2017). Finally, the Richardson et al. paper, which details the broad distribution of the sills in question, seems to be omitted from this study. It should be included and cited throughout accordingly.

L 325: As suggested here and other locations in the text, the sills are often assumed to be intruding pre-existing structures when their attitudes are oblique to the principle stresses. What about faults and fractures generated at the propagation front of sills related to uplift of the overburden (e.g., Fig 7 of Thomson 2007)? I don't think these would count as pre-existing, but would be oblique to the least compressive stress, and thus fit with the model presented. Note that they don't have to be as high-angle as the ones in the figure referred to.

Figure 1: Change normal fault regime, to "extensional regime". Change thrust fault regime to "compressional regime". We are focusing on intrusions in this paper, and not faults.

Figure 3: Same as figure 1. Change to "extensional regime", "strike-slip regime", and "compressional regime".

Also, I think in this figure there needs to be a little more explanation in the caption. From my understanding, if the pole to the fracture plane is situated in a certain color

portion of the stereonet, then the "color" indicates the opening angle ($\mu$). So if the pole of the fracture plane is in a black region, it will have an opening angle between 80-90 degrees (and subsequently a large true thickness). If the pole of the fracture plane is in a yellow regime, it will have an opening angle between 0-10 degrees, and have a smaller true thickness.

I doubt this is intuitive to most readers looking only at this figure, and I think the paper would benefit from taking the time to explain it in greater detail, perhaps using an example like I did above. [Note, I noticed later that the text discusses poles to planes, but I still think it should be explained first thing when the reader first views the caption.]

Recommend references

Richardson, J. A., Connor, C. B., Wetmore, P. H., Connor, L. J., & Gallant, E. A. (2015). Role of sills in the development of volcanic fields: Insights from lidar mapping surveys of the San Rafael Swell, Utah. Geology, 43(11), 1023-1026.

Thomson, K. (2007). Determining magma flow in sills, dykes and laccoliths and their implications for sill emplacement mechanisms. Bulletin of Volcanology, 70(2), 183-201.

---

## Referee Comment (RC2) · N. Le Corvec (Referee) · 25 Apr 2018

First of all, I enjoyed reading this paper. It is well written and to the point, I guess a lot of work has already been done to reach this quality. Figures are self-explanatory. The authors develop based on previous work (Jolly et Sanderson 1997) to define their method to better quantify the paleostress state from horizontal magma intrusions (sill). This work is based on one of the co-author recent publications (Walker 2016, Walker et al. 2017), nonetheless this methodology adds very interesting new tools to better decipher the interaction between tectonic and magmatic systems.

Major comments: Despite being short, I think the paper lacks references especially on the role of pre-existing fractures on magma intrusions, and on the mechanics of fracture

propagation. Line 328: "Failure of intact rock requires a higher fluid pressure than for reactivation of pre-existing structures". I don't agree with that. As you said yourself in the paper it depends on the magma pressure and the orientation of the fractures, as well as if it sealed or not. In addition, the propagation of the tip of a crack would tend to have a fracture to not reopen pre-exiting ones due to their angular orientations with the local stress (or maybe on short distance [Gaffney et al., 2007]).

Minor comment: Line 57: "dilation of a fluid-filled pre-existing...". I think there is a mistake here. I don't believe you mean to dilate a fracture with liquid in it?

Gaffney, E. S., B. Damjanac, and G. A. Valentine (2007), Localization of volcanic activity: 2. Effects of pre-existing structure, Earth and Planetary Science Letters, 263(3-4), 323-338.

---

## Author Comment (AC1) · 15 Jun 2018

We thank Reviewer 1 for their concise and constructive comments on the above manuscript. In almost all cases we have implemented the suggested changes. Below we present a full breakdown of the changes we have made, and our response to individual comments. The main amendments to the manuscript are:

1) Introduction: Reviewer 1 suggested to provide more detail on why sills are typically omitted from paleostress analyses. This has been added: See L47-54.

2) Mechanical model (Figure 3): Reviewer 1 suggested to provide additional explanation for how changes in the stress regime, as well as changes in fluid pressure, affect the range of fracture attitudes that would dilate. We have added a short explanation to

the manuscript to better explain this: See L155-163.

3) Discussion: The discussion has been split into sub-sections and part has been re-written in a clearer and more concise style: See L275-315.

Reply to Reviewer 1

General Comments:

1) As discussed below, there may be potential for this method to be misapplied in the absence of detailed investigations of the timing and history of intrusive events, and the authors should explicitly discuss the critical field observations needed before applying the approach. The authors have previously done extensive and careful field work in this study area (e.g., Walker et al., 2017), so I believe the approach is valid for the San Rafael field. However, the method assumes that the stress field remains constant during emplacement of the measured intrusions, and thus the magnitude of fluid pressure (i.e., magma) is the critical parameter that determines the spread of the intrusion attitude data. With this in mind, the method also requires the emplacement of the measured intrusions, which exhibit a variety of attitudes, to be closely spaced in time (i.e., L137: "created during the same dilational event"). If one were to take this approach to a different field setting, it therefore may be misapplied if the timing of events is unknown.

So with this in mind; in contrast to the model presented in this study, what if the magnitude of the fluid pressure remained relatively low (thus restricting the range of intrusion attitudes at any given time and place), but the orientation of the stress field varied both temporally and spatially? Such a scenario would seem really likely under a classic cone sheet model for example.

The authors somewhat address this point on L 267-285, but I think a little more discussion is needed so that readers can explicitly see the potential pitfalls with the approach, and be shown how to deal with these with detail field observations and measurements.
For example, what is unique about this study, compared to Jolly and Sanderson (1997), is that the authors also present the attitudes of fractures exhibiting compressional shear failure, thereby tightly bounding the ellipse in the stereonets that provide constraints on fluid pressures. The interconnectivity between intrusions of different attitudes is also illustrated, suggesting they formed near-contemporaneously. I feel that careful measurements like these robustly support the model, and future studies could also consider this approach.

Reply: These are excellent points, our statement "created during the same dilational event" implies that intrusions with different attitudes will be either mutually cross-cutting, or linked together (e.g. our Figs 1 and 7). This is comparable to conjugate or orthorhombic fault sets where the mutual cross cutting relationships signify that the structures formed within the same stress state but not exactly at the same time. A lack of mutual cross cutting, or linking, relationships between intrusive sets would indicate that they represent separate events. Relative timings should therefore be identified in the field, and different intrusive sets, which will either represent different fluid pressures and/ or different stress states, should be recorded and analysed individually.

We have added in a short section that addresses these comments: See L155-163.

2) It would be useful for the reader to also have some quick background on why sills have been omitted from these types of paleostress analyses in the introduction. This could be included as a few sentences immediately after the comments on lines 46-48.

To me, this issue stems from the problem that (1) sills in many cases must be fed by dikes, and are often observed in regions which are thought to be extensional. Thus if we consider stress as the primary control on sill orientation, then the least compressive stress must be both vertical and horizontal in the region where sills are fed, which is often in an extensional setting. The effects of mechanical layering essentially act as a work-around for this paradox. (2) Sills are often observed intruding sedimentary layers, although this is no surprise as sedimentary layering is often horizontal. I know

this group has brought up points such as these in recent papers, but I feel that it is important to bring these points to the forefront to provide broader context for the paper.

Reply: Thank you, we have added in a few lines to address this: See L47-54.

Specific Comments:

1) L 115: Please consider the terms "extensional" and "compressional" as the subject matter is intrusions rather than faults.

Reply: we see why Review 1 suggests this, however 'extensional' and 'compressional' refer to a strain and a stress, respectively; additionally these terms do not give a clear indication of principal stress axes orientation (e.g. horizontal extension and horizontal compression could both be used to describe a strike-slip regime). One of the key points of the manuscript is to show that intrusions (particularly sills) may be used as records of paleostress, in a similar fashion to faults. In the case of this study, thrust faults accommodate shortening either when there was no magmatic input, or in areas that did not become linked to a magmatic source; the sills are therefore representative of an otherwise thrust fault regime. We feel, therefore, that the association between intrusion attitudes and a given deformation regime is appropriate: L124 – 126 have been amended to thrust-fault regime ($\sigma$v = $\sigma$3), strike-slip regime ($\sigma$v = $\sigma$2), and normal-fault regime ($\sigma$v = $\sigma$1).

2) L 141-143: The other possibility is that it is multiple events where the orientation of the principle stresses has varied. I don't think this is the case in the paper, but it should not be overlooked. See Comment #1.

Reply: please see reply to comment General Comment 1.

3) L 162-163: No direct feeder relationships are observed, which is very rare in nature anyhow, but they have been inferred. For example, Figure 3 of Richardson et al. (2015) points to potential dike feeder based on sill thicknesses. Additionally, the thickness distributions of these sills do suggest potential NNE-SSE-trending feeders, which is

parallel to the dikes (i.e., Walker et al., 2017). Finally, the Richardson et al. paper, which details the broad distribution of the sills in question, seems to be omitted from this study. It should be included and cited throughout accordingly.

Reply: Richardson et al. (2015) do infer a feeding relationship between the dikes and sills (their Figs. 2 & 3), however Walker et al. (2017) demonstrate at that for the Central Cedar sill location dikes cut thin sills, and the dikes within the volcanic breccia body have chilled contacts (their Fig. 5), which suggests that at that stratigraphic level the dikes and sills did not have a 'feeding' relationship. It is possible that both inferences are correct, in that the sills may have been fed by the dikes, but that transition from dike to sill was long-lived and involved several pulses of magma, with small enough volumes and/or enough time to form cooled contacts at the margin of the conduit. Since that observation is from the North of the San Rafael area, and the models and observations in the present study are limited to the south of the area, discussion of that locality is arguably beyond the scope of the paper. We have cited Richardson et al. (2015) where appropriate.

4) L 325: As suggested here and other locations in the text, the sills are often assumed to be intruding pre-existing structures when their attitudes are oblique to the principle stresses. What about faults and fractures generated at the propagation front of sills related to uplift of the overburden (e.g., Fig 7 of Thomson 2007)? I don't think these would count as pre-existing, but would be oblique to the least compressive stress, and thus fit with the model presented. Note that they don't have to be as high-angle as the ones in the figure referred to.

Reply: This is a very good point, but we feel that it is effectively unknowable with the current field data. At present it is not possible to constrain whether these are truly "existing" fractures, or whether they are faults/fractures formed ahead of the propagating tip. Further detailed analysis of fracture populations around the intrusions, at the regional and local scale, may provide some useful information as to whether the fracture attitudes are local to sills, but from our existing data set, this appears not to

be the case, and further, it may not be possible to distinguish different and super-imposed populations anyway. Analysis of fracture/fault slip distributions ahead of sill tips may give some further insights to the problem, assuming that the fractures have distinguishable slip distribution profiles depending on whether they are pre-existing or newly-generated. Distinguishing displacement maxima and/or slip distribution profiles for dykes may be possible because of the potential abundance of offset markers, and likewise for steep faults/fractures, as with those shown in Thomson (2007). In a field study of sills, this would require excavation of the fracture sets; we have not done this, and it is unclear whether this information would be preserved at any useable scale and quantity owing to the gentle dips of the fractures, and the lack of offset markers.

We have added an additional section which describes the conditions necessary for dilation of high-angle fractures: See L138-152.

5) Figure 1: Change normal fault regime, to "extensional regime". Change thrust fault regime to "compressional regime". We are focusing on intrusions in this paper, and not faults.

6) Figure 3: Same as figure 1. Change to "extensional regime", "strike-slip regime", and "compressional regime". Also, I think in this figure there needs to be a little more explanation in the caption.

From my understanding, if the pole to the fracture plane is situated in a certain color portion of the stereonet, then the "color" indicates the opening angle (_). So if the pole of the fracture plane is in a black region, it will have an opening angle between 80-90 degrees (and subsequently a large true thickness). If the pole of the fracture plane is in a yellow regime, it will have an opening angle between 0-10 degrees, and have a smaller true thickness.

I doubt this is intuitive to most readers looking only at this figure, and I think the paper would benefit from taking the time to explain it in greater detail, perhaps using an example like I did above. [Note, I noticed later that the text discusses poles to planes,

but I still think it should be explained first thing when the reader first views the caption.]

Reply to comments 5 & 6: Please see reply to Specific Comment (1). The caption of Fig. 3 has been expanded to explain the relationship between the colour contouring, opening angle, and intrusion thickness.

Recommend references

Richardson, J. A., Connor, C. B., Wetmore, P. H., Connor, L. J., & Gallant, E. A. (2015). Role of sills in the development of volcanic fields: Insights from lidar mapping surveys of the San Rafael Swell, Utah. Geology, 43(11), 1023-1026.

Thomson, K. (2007). Determining magma flow in sills, dykes and laccoliths and their implications for sill emplacement mechanisms. Bulletin of Volcanology, 70(2), 183-201.

Reply: Thank you for the suggestions, Richardsson et al. (2015) has been cited where appropriate. We have used an alternative paper (Malthe-Sørenssen et al., 2004) to refer to the emplacement of saucer-shaped sills.

Please also note the supplement to this comment:
https://www.solid-earth-discuss.net/se-2018-17/se-2018-17-AC1-supplement.pdf

**Supplement:**

[revised manuscript text omitted]

---

## Author Comment (AC2) · 15 Jun 2018

We thank Nicolas (Reviewer 2) for his concise and constructive comments on the above manuscript. Below we present a full breakdown of the changes we have made, and our response to individual comments. Based on the comments, the main amendments to the manuscript are:

1) Fracture Mechanics: Reviewer 2 suggested that more explanation was required for the effect of pre-existing fractures on hydrofracture propagation. We have added a short section to address how pre-existing fractures that are oriented at a high angle to $\sigma 3$ can influence hydrofracture propagation: See L138-152.

Reply to Reviewer 2

[Figure]

Major comments: Despite being short, I think the paper lacks references especially on the role of pre-existing fractures on magma intrusions, and on the mechanics of fracture propagation. Line 328: "Failure of intact rock requires a higher fluid pressure than for reactivation of pre-existing structures". I don't agree with that. As you said yourself in the paper it depends on the magma pressure and the orientation of the fractures, as well as if it sealed or not. In addition, the propagation of the tip of a crack would tend to have a fracture to not reopen pre-exiting ones due to their angular orientations with the local stress (or maybe on short distance [Gaffney et al., 2007]).

Reply: Thank you for pointing this out, we agree and Lines 328 – 333 have been removed. Additionally, we understand why Reviewer 2 suggests to include fracture mechanics literature, however the model proposed here does not refer to hydrofracture initiation or propagation; it simply calculates the opening angle for fractures that were linked to the fluid system. We have added in a short section to address how pre-existing fractures that are oriented at a high angle to $\sigma 3$ can influence hydrofracture propagation: See L138-152.

Additional references have also been added into the paper regarding the geometry of extensional-shear fractures formed via failure of intact rock: Hancock, 1985; Sibson, 1996; and Ramsey and Chester, 2004.

Minor comment: Line 57: "dilation of a fluid-filled pre-existing: : :". I think there is a mistake here. I don't believe you mean to dilate a fracture with liquid in it?

Reply: Correct: "fluid-filled" has been changed to "cohesionless".

Recommend references

Gaffney, E. S., B. Damjanac, and G. A. Valentine (2007), Localization of volcanic activity: 2. Effects of pre-existing structure, Earth and Planetary Science Letters, 263(3-4), 323-338.

Reply: Thank you for the suggestion, the reference has been cited where appropriate.

Please note: A tracked-changes version of the manuscript has been added as a sup-
plement to the reply to Reviewer 1.